# Cytokine Gene Expression Profiles during HIV and Helminth Coinfection in Underprivileged Peri-Urban South African Adults

**DOI:** 10.3390/diagnostics13152475

**Published:** 2023-07-25

**Authors:** Miranda N. Mpaka-Mbatha, Pragalathan Naidoo, Khethiwe N. Bhengu, Md. Mazharul Islam, Ravesh Singh, Nomzamo Nembe-Mafa, Zilungile L. Mkhize-Kwitshana

**Affiliations:** 1Department of Medical Microbiology, School of Laboratory Medicine and Medical Sciences, College of Health Sciences, Nelson R. Mandela Medical School Campus, University of KwaZulu-Natal, Durban 4001, South Africa; pragalathan.naidoo@gmail.com (P.N.); bhengukn@mut.ac.za (K.N.B.); nomzamo.nembe@gmail.com (N.N.-M.); mkhizekwitshanaz@ukzn.ac.za (Z.L.M.-K.); 2Division of Research Capacity Development (RCD), South African Medical Research Council (SAMRC), Tygerberg, Cape Town 7505, South Africa; 3Department of Biomedical Sciences, Faculty of Natural Sciences, Mangosuthu University of Technology, Umlazi, Durban 4031, South Africa; 4Department of Animal Resources, Ministry of Municipality, Doha P.O. Box 3508, Qatar; walidbdvet@gmail.com; 5Department of Medical Microbiology, School of Laboratory Medicine and Medical Sciences, College of Health Sciences, Howard College, University of KwaZulu-Natal, Durban 4041, South Africa; singhra@ukzn.ac.za

**Keywords:** HIV, helminths, coinfection, immune response profiles, inflammation

## Abstract

Background: Intestinal helminth parasites are potent stimulators of T helper type 2 (Th2) and regulatory Th3 anti-inflammatory immune responses, while human immunodeficiency virus (HIV) infections are activators of predominantly T helper type 1(Th1) pro-inflammatory responses. Studies investigating the immune profiles of individuals coinfected with helminths and HIV are scarce. Although it is well known that helminths cause a type 2 immune response during the chronic stage of infection that is characterised by Th2 cell differentiation, eosinophil recruitment, and alternative macrophage activation, the immune mechanisms that regulate tissue damage at the time of parasite invasion are poorly understood. Aim: The aim of the study was to determine the cytokine gene expression profiles during HIV and helminth coinfection in underprivileged South African adults living in a peri-urban area with poor sanitary conditions and a lack of clean water supply. Method: Study participants (*n* = 164) were subdivided into uninfected controls, HIV-infected, helminth-infected, and HIV and helminth-coinfected groups. The Kato–Katz and Mini Parasep techniques and *Ascaris lumbricoides*-specific Immunoglobulin E (IgE) and Immunoglobulin G4 (IgG4) levels were used to detect helminth infections. Participants’ HIV status was determined using two HIV1/2 antibody test kits. RNA was isolated from white blood cells for cytokine (Th1-, Th2-, and Th17-related) and transcription factor gene expression profiling using real-time PCR. Results: Multivariate regression data were adjusted for age, gender, BMI, antiretroviral treatment (ART), and nutritional supplement intake. The HIV and helminth-coinfected group had significantly higher tumour necrosis factor alpha (*TNF-α*) (adjusted β = 0.53, *p* = 0.036), interleukin 2 *(IL-2*) (adjusted β = 6.48, *p* = 0.008), and interleukin 17 (*IL-17*) (adjusted β = 1.16, *p* = 0.001) levels and lower GATA binding protein 3 *(GATA3*) levels (adjusted β = −0.77, *p* = 0.018) compared to the uninfected controls. No statistical significance was noted for Th2-related cytokines. Conclusion: The coinfected group had higher proinflammatory Th1- and Th17-related cytokine gene expression profiles compared to the uninfected controls. The findings suggest that pro-inflammatory responses are elevated during coinfection, which supports the hypothesis that helminths have a deleterious effect on HIV immune responses.

## 1. Introduction

In sub-Saharan Africa (SSA), there is extensive epidemiological overlap between human immunodeficiency virus/acquired immune deficiency syndrome (HIV/AIDS) and intestinal helminth infections [1,2,3,4], As diseases of poverty, it is likely that an individual infected with HIV may also be infected with one or more parasites and vice versa. Both these infections are highly prevalent in individuals from poor demographic and socioeconomic backgrounds, with helminth infections being predominant in individuals exposed to unsanitary living conditions and a lack of clean water supply [5,6].

Chronic helminth infections may result in an immunomodulatory response in HIV- and helminth-infected individuals, which increases parasite survival [7]. Helminth infection may lead to the increased expression of the co-receptors of HIV-1 chemokines on T-lymphocytes (T cell) and monocytes [8,9], facilitating easy HIV-1 entry and increasing the pool of cells susceptible to infection. Peripheral blood mononuclear cells (PBMCs) obtained from helminth-infected individuals showed increased susceptibility to HIV-1 infection [10]. Both HIV and helminth infections lead to the persistent activation of the immune system and, as a result, an impairment of host immune responses becomes evident [11,12].

T helper type 1 (Th1) proinflammatory immune responses such as tumour necrosis factor alpha (*TNF-α*), interferon gamma (*IFN-γ*), and interleukin (*IL-2*) play an important role in the host’s defence against HIV [13], especially during the acute phase. On the contrary, the T helper type 2 (Th2) anti-inflammatory response has been demonstrated to be vital for the clearance of helminthic parasites [11]. Host immune response to helminth infection is associated with the production of *IL-4, IL-5, IL-9, IL-10,* and *IL-13* and, subsequently, the development of a strong immunoglobulin E (IgE) response [14,15]. A marked shift of the T cell Th1 response to a Th2 response has been suggested to lead to increased HIV transmission, impaired protection against HIV, elevated plasma HIV RNA levels, and rapid disease progression among coinfected individuals in SSA [3]. In addition, it has been shown that immune activation is significantly reduced during antiretroviral therapy (ART) and helminth parasite treatment [16]. Furthermore, regulatory T cells (Tregs) are induced, resulting in the production of *IL-10* and transforming growth factor beta (*TGF-β*) [15]. IL-10 is a regulatory cytokine that controls inflammation [7].

Regulatory cluster of differentiation (CD) 4^+^ T cells express CD25 (*IL-2Rα*) and co-express the transcription factor forkhead box P3 (*FoxP3*) [13,17,18,19]. These markers are used to identify and characterise Tregs and have been reported in HIV and helminth infections separately [20].

Tregs are essential for the maintenance of self-tolerance, and they can suppress the activation, proliferation, and effector functions of a wide range of immune cells, including CD4^+^ and CD8^+^ T cells [20]. Little is known about the impact of Tregs on HIV-infected individuals. However, during helminth infection, high levels of Treg cells are induced, which become potential targets for HIV acquisition [19,20]. Once these cells become infected, they can potentially contribute to HIV disease progression in HIV and helminth-coinfected individuals. Other studies have found that *FoxP3*-expressing T cells tend to accumulate in the lymphoid tissues of people living with HIV [21,22].

In South Africa (SA), the majority of people are infected with one or more intestinal helminth parasites (*Ascaris lumbricoides*, *Trichuris trichiura*, *Schistosoma mansoni*, *Ancylostoma duodenale*, and *Necator americanus*), making them more susceptible to acquiring a predominant anti-inflammatory Th2 immune response [23,24]. As a consequence, this scenario could dampen the production and expression of vital pro-inflammatory Th1 cytokines that are crucial for controlling early-stage HIV pathogenesis and progression. There is a lack of data that conclusively demonstrate whether these interactions are more detrimental or beneficial for the host when compared to singly infected individuals. In KwaZulu-Natal (KZN), one of nine provinces in SA, there is a high degree of poverty, which is associated with high rates of HIV and helminth infections and malnutrition. Taking all this into consideration, we aimed to determine the immune response profiles during HIV and helminth coinfection in underprivileged SA adults residing in a peri-urban area of KZN.

## 2. Methods

### 2.1. Study Design, Setting and Patient Recruitment 

A descriptive cross-sectional study was conducted between March 2020 and May 2021 at five clinics located in a peri-urban area within the eThekwini district, South of Durban, in the KwaZulu-Natal province of South Africa. Essentially, participants (414) recruited into the study were adult (≥18 years) clinic attendees in a primary health care setting that also provides HIV testing and counselling. Some were accompanying patients to the clinic while others were either clinic employees or community members who had been informed about the study. The very sick and frail, pregnant females, those with known infections including chronic diseases or those on medication, or those using illicit drugs or excessive alcohol were excluded. Detailed information about the participants, including the inclusion and exclusion processes, can be found in a previously published article [25]. From the overall sample, a sub-sample of 164 participants was selected randomly for cytokine and transcription factor gene expression studies in the current work. 

### 2.2. Stool Collection and Helminth Parasite Detection

Two sets of stool samples were collected from participants over a two-day period to enhance the sensitivity and specificity of coproscopy. The Kato–Katz and the modified formal ether Mini Parasep^®^ SF faecal parasite concentrator techniques were used to detect helminth parasites (eggs and worms). A comprehensive description of these procedures can be found in previous publications [25,26].

### 2.3. Blood Collection, Biochemical Analysis and HIV Confirmation

Whole blood was collected from all participants using the Vacuette^®^ EDTA and serum blood tubes and sent to diagnostics laboratories for biochemical analysis, CD4 counts, HIV-1 viral loads, and full blood counts (FBC). Blood serum was used for total IgE and Ascaris lumbricoides-specific IgE and IgG4, and HIV status confirmation using the Alere Determine^TM^ HIV-1/2 Ag/Ab Combo rapid test (Orgenics Ltd., Yavne, Israel) was used to further validate the participant’s HIV status. Inconclusive results were confirmed by using the ICT HIV-1/2 Ag/Ab test kit (ICT diagnostics, Cape Town) [25,26].

### 2.4. Classification of Study Participants

The population was classified as follows: (i) uninfected controls (*n* = 20), (ii) HIV-infected (*n* = 60), (iii) helminth-infected (*n* = 37), and (iv) HIV and helminth-coinfected groups (*n* = 47). 

### 2.5. RNA Isolation and cDNA Synthesis

Briefly, 5 mL of whole blood samples collected in EDTA blood tubes were lysed using the red blood cell (RBC) lysis buffer (Invitrogen, Waltham, MA, USA). Thereafter, the white blood cells were collected and stored in TRIzol^®^ reagent (Invitrogen) and kept at −80 °C until use for RNA extraction. The PureLink^TM^ RNA Mini Kit (ThermoFisher Scientific, Waltham, MA, USA, cat no. 1218305) and PureLink^TM^ DNase Set (ThermoFisher Scientific, cat no. 12185010) were used to isolate RNA from white blood cells that were stored in TRIzol.

The NanoDrop 2000 spectrophotometer (ThermoFisher Scientific) was used to assess the concentration and purity of the isolated RNA. RNA samples with an Abs260nm/Abs280nm ratio value (an indicator of protein contamination) between 1.8 and 2.0 were considered to be pure. Samples with an Abs260nm/Abs230nm ratio value (an indicator of solvent contamination) between 1.8 and 2.2 were considered acceptable.

The standardised RNA samples (1 µg) were converted to cDNA using the High-Capacity cDNA Reverse Transcription Kits (ThermoFisher Scientific, cat no. 4374966). The total cDNA concentration was quantified and samples were used only if the optical density at 260 nm (Abs260/Abs280) was >1.8. Thereafter, the cDNA samples were stored at −80 °C until further use for gene expression studies.

### 2.6. Gene Expression Analysis by Real Time-Quantitative PCR (RT-qPCR)

Real Time-Quantitative PCR (RT-qPCR) was used to determine the levels of cytokine gene expression (*Th1, Th2, Th17*, and transcription factors). The master mix for assays was prepared by adding (i) 2.5 µL PCR-grade water (ThermoFisher Scientific); (ii) 0.25 µL FAM-labelled of each probe/primer mix (*Glyceraldehyde 3-phosphate dehydrogenase* (GAPDH) (Hs99999905_m1)), interferon-gamma (*INF-γ*) (Hs00989291_m1), tumour necrosis factor-alpha (*TNFα*) (Hs00174128_m1), nuclear factor of activated T-cells 2 (*NFATC2*) (Hs00905451_m1), *Granzyme B* (Hs00188051_m1), perforin (Hs00169473_m1), Eomesodermin (*Eomes*) (Hs00172872_m1), GATA-binding protein 3 (*GATA3*) (Hs00231122_m1), interleukin-17 (*IL-17*) (Hs01056316_m1), transforming growth factor-beta (*TGF-β*) (Hs00234244_m1), and forkhead box P3 (*FoxP3*) (Hs01085834_m1) (ThermoFisher Scientific); (iii) 1.25 µL Fast Start 4× probe master mix (ThermoFisher Scientific); and (iv) 1 µL cDNA to make a total volume of 5 µL per sample. 

Amplification was performed at 95 °C for 30 s followed by 45 cycles comprising denaturation at 95 °C for 3 s and annealing at 60°C for 30 s and extension at 72 °C for 30 s. All reactions were run in duplicate.

Serial dilutions of pooled cDNA from total RNA were performed for each target gene and reference gene. These dilutions served as standard curves for quantitative analysis, ranging from 1 ng to 1000 ng.

Data were collected using the Applied Biosystems QuantStudio 5 V.2.3 software (ThermoFisher Scientific). The glyceraldehyde 3-phosphate dehydrogenase (GAPDH) gene was used as a reference gene. The results are depicted as a ratio of the gene of interest over GAPDH. The levels of *Th1* (*IFN-γ, TNF-α, IL2, perforin, and Granzyme B*), *Th17* (*IL-17*), and *Th2* (*IL4, IL10*) (*TGF-β*)-related cytokine gene expression and transcription factors (*Foxp3, GATA3, NFATC2*, and *Eomes*) were compared among the study participants classified as (i) uninfected controls, (ii) HIV-infected only, (iii) helminth-infected only, and (iv) HIV and helminth-coinfected groups. 

### 2.7. Statistical Analysis

All statistical analysis was performed by using Stata Statistical Software: Release 17 (College Station, TX, USA: StataCorp LLC. StataCorp. 2019). The data of demographic and clinical characteristics, except for gender, were analysed using the Kruskal–Wallis test and Dunn’s multiple-comparison test, and the results are presented as the median (25th–75th percentiles). Data for gender are presented as n (%) and were analysed using the Chi-squared test. Multivariate regression analysis was conducted for the data for immune-related gene-expression, and the results are presented as unstandardised beta coefficient (β) values with their corresponding 95% confidence intervals (CIs). The uninfected control group served as the reference group for this analysis. The data were analysed both before (data in row A) and after (data in row B) adjusting for confounding variables, including age, gender, body-mass index (BMI), antiretroviral treatment (ART), and intake of vitamin and nutrient supplements. A *p*-value <0.05 was considered statistically significant.

## 3. Results

### 3.1. Demographics and Clinical Characteristics

The demographic and clinical profiles of the study participants are shown in Table 1. Out of the 164 participants in the study, 47 (28.7%) were coinfected, 60 (36.6%) were HIV singly infected, and 37 (22.6 %) were helminth singly infected. The median age and BMI were similar between the different groups. Most of the participants were female. As expected, the median CD4 counts were significantly lower in the HIV-infected and coinfected groups (*p* < 0.0001). No significant changes in eosinophils, neutrophils, lymphocytes, monocytes, basophils, and CRP were noted between the different groups. 

### 3.2. Parasite Prevalence

Table 2 highlights the species of helminths and protozoans (*Entamoeba coli* (10.40%)) identified by eggs/ova/cysts. The prevalent species were *Ascaris lumbricoides* (39.6%), *Taenia* spp. (4.3%), Schistosoma spp. (4.9%), *Strongyloides* spp. (2.4%), *Trichuris trichiura* (1.8%), *Enterobius vermicularis* (1.8%), hookworms (0.6%), and *Hymenolepis* spp. (0.90%) (Table 2).

### 3.3. Cytokine Gene Expression Profiling

The multivariate association of immune-related gene expression during HIV and helminth single infection and coinfection are presented in Table 3. Data were adjusted for age, gender, BMI, and nutrient supplement intake. In addition, the cytokine gene expression profiles were also investigated in the HIV singly infected group and the HIV and helminth-coinfected group, which were stratified based on ART therapy to determine whether ART therapy was a confounding factor and could have skewed the results. We found a significant difference for *Eomes, IL17*, and perforin in the HIV singly infected group when compared to the uninfected controls.

(i) Th1-related cytokine genes: The HIV and helminth-coinfected group had significantly higher TNF-α (adjusted β = 0.53, *p* = 0.036) and IL-2 (adjusted β = 6.48, *p* = 0.008) levels compared to the uninfected controls. (ii) Th2 and T-regulatory cytokine genes: The helminth-infected group had significantly lower TGF-β levels compared to the uninfected controls (adjusted β = −0.86, *p* = 0.033). No statistically significant difference was noted for IL-4 and IL-10. (iii) Th17-related cytokine gene: The HIV-infected (adjusted β = 1.10, *p* = 0.000), helminth-infected (adjusted β = 0.88, *p* = 0.001), and coinfected (adjusted β = 1.16, *p* = 0.001) groups had significantly higher IL-17 levels compared to the uninfected controls. (iv) Transcription factors: In comparison to the uninfected control group, the HIV-infected (adjusted β = −0.08, *p* = 0.010) and coinfected (adjusted β = −0.77, *p* = 0.018) groups had significantly lower *GATA3* levels. In addition, the helminth-infected group had significantly lower Eomes levels (adjusted β = −0.24, *p* = 0.018) compared to the uninfected controls. No statistically significant differences were noted for *FOXP3, NFATC2, Granzyme B, and perforin.*

## 4. Discussion

The current study aimed to assess the expression of different classes of cytokines during HIV and helminth coinfection in underprivileged South African adults residing in a peri-urban, poorly resourced area. Our results showed that the coinfected group had higher proinflammatory *Th1* (*TNF-α* and *IL-2*) and *Th17 (IL-17*)-related cytokine gene expression profiles and lower *GATA3* expression levels than uninfected controls. The overall findings suggest that pro-inflammatory responses are elevated during coinfection.

Upon a search of the literature, studies investigating these pro-inflammatory cytokines during coinfection were not found. However, the early stages of helminth infection promote an inflammatory response [26,27]. Likewise, during HIV infection, proinflammatory cytokines are upregulated [28,29]. Therefore, the findings of increased pro-inflammatory cytokines during HIV and helminth coinfection in the current study is in keeping with the previously mentioned studies. In the current study, however, it is not clear whether the infections with helminths were in the acute or chronic stages of infection. Although there was no statistically significant difference in *INF-γ*, typically it is elevated during HIV infection [30].

*IL-2* is essential for the initiation and resolution of inflammatory immune responses. Some studies have reported lower levels of *IL-2* during uncontrolled HIV [29], while others reported increased *IL-2* in asymptomatic donors [31]. It is, however, unclear whether these findings are a result of a variation in plasma *IL-2* during the course of HIV-1 disease or if RNA gene expression is more specific in detecting the inflammatory cytokine. The current study used RNA to determine the *IL-2* gene expression.

It is expected that a heavy burden of helminth infection results in diminished IL-2 and interferon-γ (*IFN-γ*) responses. This downregulation is caused by immune regulation, which subsequently suppresses the Th1 immune response. In addition, the current study found decreased *GATA3*. This is contrary to the reported findings that *GATA3* is most abundantly expressed in T lymphocytes, a cellular target for human immunodeficiency virus type 1 (HIV-1) infection and replication [32].

Spurious findings were noted for Th2 cytokines, whereas others reported evidence of a Th2/Treg-predominant helminth-induced response [11,26] as well as a Th1–Th2 switch, and increased Tregs during chronic HIV infection suggest that dual HIV and helminth coinfection may have an additive effect on the immune response profile. Likewise, IL-10 showed no significant difference, but *TGF-β* was upregulated, which is in keeping with the hypothesis that infections with helminth are associated with upregulated Th2 and Treg immune responses [26].

At the time of planning the study, the WHO strategy of “test and treat” (2016) had not been implemented in South Africa. The study plan was to recruit only antiretroviral therapy-naïve participants. However, by the time full ethics approval was obtained, SA had initiated the strategy. Thus, during recruitment, 131 of the 421 were on ART (and 56 of the 164 in the substudy were on ART). Inadvertently, it was expected that the antiretroviral therapy would confound the results. However, upon analysis, the results did not change when adjusted for ART. One possible explanation could be that some of the participants had only recently started antiretroviral treatment and thus it had not had an effect that significantly influenced the results. It could not be ascertained when the ART was initiated, therefore the treatment duration could not be established.

Helminth parasites resist immune expulsion via sophisticated evasion mechanisms which include the activation of host immunosuppressive regulatory T (Treg) cells [33]. One of the ways they do this is by producing a cytokine-like protein (*TGF-β*) capable of exploiting an endogenous immunoregulation pathway in the host. Since helminth infections overlap with HIV infection in areas where both infections are highly prevalent, inflammation in dually infected individuals could harm the host. It is therefore important to stress the value of the parasitological examination of these neglected diseases early to prevent chronic infections, which have devastating effects on children and HIV-infected individuals. Microscopic examination is the most simple and cost-effective screening method for the identification of medically important parasites.

One of the limitations of the study is the cross-sectional design. due to cost implications, there was no follow-up arm. The study is also limited by selection bias. Most of the participants were recruited from clinics, even though some of the controls were recruited from the community and people accompanying the patients. In particular, the selection of clinics was biased towards those with HIV Counselling and Testing (HCT) clinics. This is reflected in the high overall prevalence of HIV in the study sample. Due to the high cost of gene expression, funding was also constrained.

## 5. Conclusions

The current study showed that HIV and helminth-coinfected individuals might be more susceptible to inflammatory immune responses than those who are singly infected with either HIV or helminth, suggesting that helminths can further promote inflammation in HIV-infected individuals. The data from this study indicate that integrating deworming treatment into HIV management could be important in improving HIV disease outcomes.

## Figures and Tables

**Table 1 diagnostics-13-02475-t001:** Demographics and clinical characteristics of study participants (*N* = 164).

Parameters	Uninfected Controls (*n* = 20)	HIV-Infected Only (*n* = 60)	Helminth-Infected Only (*n* = 37)	HIV + Helminth Coinfection (*n* = 47)	*p*-Value
**Age**	56.00 (18.00–70.00)	41.00 (22.00–61.00)	35.00 (19.00–74.00)	41.00 (18.00–63.00)	0.09
**Gender**					0.13
**Males**	5 (25)	27 (45)	15 (40.5)	12 (25.5)	
**Females**	15 (75)	33 (55)	22 (29.5)	35 (74.5)	
**BMI**	26.00 (15.70–52.10)	26.40 (15.00–47.00)	25.00 (17.00–46.00)	24.80 (12.20–49.80)	0.75
**CD4 counts**	895.00 (292.00–1989)	490.00 (55.00–1255)	861.00 (359.00–1302)	553.00 (37.00–1458)	<0.01
**Viral load**		255.00 (31.00–1,500,000)		250.00 (28.00–240,000)	0.76
**Eosinophils**	0.14 (0.00–0.61)	0.10 (0.00–1.30)	0.12 (0.02–1.40)	0.12 (0.00–0.31)	0.31
**Neutrophils**	3.20 (1.52–6.71)	2.44 (0.87–28.30)	3.36 (1.40–8.70)	2.70 (0.87–5.13)	0.07
**Lymphocytes**	2.02 (0.62–3.77)	1.73 (0.57–4.50)	2.10 (0.84–3.51)	1.80 (0.74–4.73)	0.09
**Monocytes**	0.44 (0.20–0.76)	0.41 (0.16–1.34)	0.46 (0.16–0.98)	0.38 (0.21–0.87)	0.17
**Basophils**	0.03 (0.01–0.09)	0.02 (0.01–0.40)	0.03 (0.01–0.10)	0.02 (0.01–0.10)	0.17
**C-reactive protein (*n* = 135)**	5.30 (1.10–25.20)	7.10 (0.78–95.60)	4.30 (0.60–114.60)	3.90 (0.50–46.10)	0.23

Parameters are expressed as medians in count × 10^9^/L, age is expressed in years, gender is expressed in *n* (%), BMI in kg/m^2^, viral load in copies per mL of blood, and C-reactive protein in mg/L.

**Table 2 diagnostics-13-02475-t002:** Parasite prevalence in the study population (*n* =164).

Parasite Species	*n*	%
*Ascaris lumbricoides*	65	39.6
*Taenia* spp.	7	4.3
*Schistosoma* spp.	8	4.9
*Strongyloides* spp.	4	2.4
*Trichuris trichiura*	3	1.8
Hookworms	1	0.6
*Enterobius vermicularis*	3	1.8
*Entamoeba coli*	11	6.7
*Hymenolepis* spp.	1	0.6

**Table 3 diagnostics-13-02475-t003:** Multivariate association of immune-related gene-expression during HIV and helminth single infection and coinfection.

	Unstandardised β-Coefficient Values (Reference Group: Uninfected Controls)						
Parameters	HIV-Infected			Helminth-Infected		HIV and Helminth-Coinfected	
	β (95% CI)		*p*	β (95% CI)	*p*	β (95% CI)	*p*
T-helper type 1 (Th1) and Transcription Factors:							
Tumour necrosis factor-α (TNF-α)	A	0.18 (−0.35–0.71)	0.510	0.11 (0.33–0.55)	0.607	0.52 (0.09–0.95)	0.020
	B	0.31 (−0.66–1.29)	0.511	0.13 (−0.32–0.58)	0.559	0.61 (−0.21–1.42)	0.134
Interferon-gamma	A	0.06 (−0.28–0.40)	0.734	−0.11 (−0.465–0.24)	0.513	0.16 (−0.16–0.47)	0.123
(IFN-γ)	B	0.03 (−0.62–0.63)	0.992	−0.08 (−0.48–0.30)	0.659	−0.17 (−0.66–-0.31)	0.463
Interleukin 2 (IL2)	A	1.80 (−6.44–10.05)	0.660	2.55 (−2.15–7.25)	0.279	5.14 (1.21–9.07)	0.012
	B	4.65 (−7.84–17.14)	0.448	3.60 (−1.94–9.08)	0.196	7.37 (0.78–13.96)	0.030
Nuclear factor of activated T cells 2 (NFATC2)	A	−0.22 (−0.77–0.32)	0.420	−0.53 (−1.04–−0.01)	0.044	−0.02 (−0.72–0.68)	0.947
	B	−0.07 (−0.68–0.57)	0.801	−0.53 (−1.10–0.39)	0.067	−0.13 (−0.71–0.93)	0.673
GATA3	A	−0.76 (−1.34–0.20)	0.012	−0.49 (−1.10)	0.106	0.16 (−0.16–0.47)	0.322
	B	−0.84 (−1.94–0.23)	0.128	−0.59 (−1.22–044)	0.067	−0.60 (−1.67–0.48)	0.258
Eomesoderin (Eomes)	A	−0.24 (−0.76–0.28)	0.364	0.48 (−0.93–0.04)	0.035	−0.38 (−0.94)	0.167
	B	−1.03 (−1.82–0.23)	0.014	−0.24 (−067–0.19)	0.018	−0.83 (−1.74–0.08)	0.072
T-helper type 17 (Th17)							
Interleukin-17 (IL-17)	A	1.21 (0.79–1.64)	0.000	0.78 (0.31–1.25)	0.002	1.13 (0.58–1.67)	0.000
	B	0.81 (0.01–1.61)	0.048	0.88 (0.39–1.37)	0.001	01.25 (0.45–2.06)	0.004
Immune Protein and Proteases							
Granzyme B (GrB)	A	0.05 (−0.32–0.42)	0.794	−0.33 (−074–0.80)	0.112	0.09 (−0.28–0.47)	0.616
	B	−0.21 (−0.83–0.42)	0.497	−0.24 (−0.67)	0.262	−0.18 (−0.70–0.34)	0.475
Perforin	A	−0.79 (−0.60–0.19)	0.309	−0.27 (−066–0.129)	0.180	−0.29 (−0.65–0.08)	0.123
	B	−0.72 (−1.33–0.12)	0.022	−0.36 (−0.78–0.60)	0.091	−0.36(−1.00–0.29)	0.263
T-helper type 2 (Th2) and Transcription factors:							
Interleukin 4 (IL4)	A	−0.16 (−6.44–10.04)	0.455	0.08 (−0.38–0.55)	0.718	−0.04 (−0.55–0.47)	0.883
	B	−0.08 (−0.64–0.48)	0.757	0.05 (−0.50–0.58)	0.856	0.08 (−0.47–0.63)	0.760
Forkhead box P3 (FOXP3)	A	0.26 (−0.27–0.79)	0.334	0.20 (−0.21–0.60)	0.339	0.10 (−0.4–0.61)	0.682
	B	0.88 (−0.24–2.00)	0.118	0.26 (−0.18–0.70)	0.234	0.18 (−0.72–1.09)	0.675
Regulatory							
Interleukin 10 (IL10)	A	−0.10 (−0.54–0.42)	0.803	−0.152 (0.57–0.26)	0.463	−0.09 (−0.52–0.046)	0.685
	B	−0.05 (−0.65–0.55)	0.867	0.00 (−0.46–0.47)	0.985	−0.05 (−0.63–0.53)	0.867
Transforming growth factor-β (TGF-β)	A	−0.50 (−1.20–0.20)	0.158	−1.04 (−1.80–−0.27)	0.009	−0.68 (−1.55–0.19)	0.119
	B	−1.05 (−2.42–0.33)	0.128	−0.86 (−1.65–−0.07)	0.033	−0.86 (−2.39–0.68)	0.255

A: Unadjusted data. B: Data were adjusted for age, gender, BMI, ART, vitamin and nutrient supplements intake.

## Data Availability

All data are available on request from the corresponding author.

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
