# Peer review of "Cytokine Gene Expression Profiles during HIV and Helminth Coinfection in Underprivileged Peri-Urban South African Adults"

_diagnostics, 2023, doi:10.3390/diagnostics13152475_

Round 1

Reviewer 1 Report

Please go through the comments posted on the paper and make the needful changes. Some major gaps identified in the paper are:

1. The inclusion and exclusion criteria of the participant selection for the study is not well explained. Infection alone cannot be considered as a selection criterion. many comorbidities may be present as a result of/ independent of the infection that may cause a change a cytokine gene expression. Therefore proper selection criterion is critical for unbiased results. 

2. The participation of HIV infected individuals in ART is not mentioned in the paper. The type of therapy undergoing by the participants (if any) can dramatically affect the results. besides, if only some individuals are on ART in each group, that might skew the results too.

3. The statistical methods are not well explained. please explain in detail the statistical methods used, especially for multivariate analysis.

4. Why did the authors not consider using ELISA to test the cytokine levels in parallel to the gene expression profiling? The rationale and the merit of gene expression profiling were not mentioned and the novelty is not apparent.

5. Including a limitations section explaining all possible shortcomings of the study is recommended.

Author Response

Dear reviewer

Thank you for the feedback, your comments have been attended to and the responses are attached below.

Regards,

Mpaka-Mbatha

Reviewer 2 Report

Dear Editor, thank you for the opportunity to review the manuscript entitled:Cytokine gene expression profiles during HIV and helminth confection in underprivileged peri-urban South African adults"by M. Mpaka.

The focus of the manuscript is interesting, original and well fit with the journal; 

I suggest to Author to improve the description of enrolled patients:

1) HIV patients assumed antiretroviral therapy?

2) If yes, why the levels of HIV-RNA were detectable? May the adherence was sub optimal or the time in therapy was short?

3) Can the Author describe if patients presented symptoms for the infection by helminth?

4) In the discussion the Author should stress the importance to do parasitological exam of the stool.

Author Response

Dear Reviewer

Thank you for your feedback, your comments have been attended to and the responses are attached below.

Regards,

Mpaka- Mbatha

Round 2

Reviewer 1 Report

After addressing the comments, the quality of the manuscript has been significantly improved.